# Efficient and Effective Dynamic Hair Reconstruction via Strand Gaussians

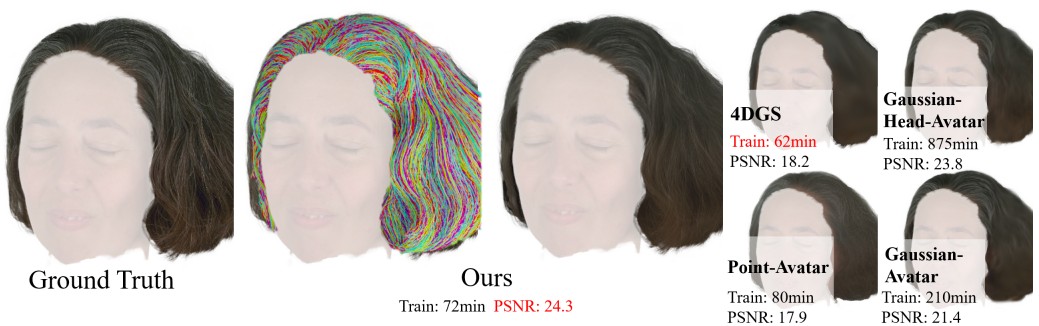

Figure 1: We introduce an effective and efficient dynamic hair reconstruction method via strand gaussians. Specifically, we made several modifications to promote sparsity-driven efficient optimization approaches. Given multi-view videos, our algorithm faithfully reconstructs the hair strand structure and appearance, achieving photorealistic rendering quality that is comparable to state-of-the-art methods such as 4DGS and Gaussian-Avatar, while enjoying significant speed-up.

## Abstract

Reconstructing dynamic human hair is a crucial way to acquire high quality 3D assets that empowers downstream tasks such as human avatars and animation. Recently, several research works propose using strand gaussians to model human hair, leading to superior reconstruction quality. However, their optimization algorithms admit significant computational burden and typically take hours to days reconstruct dynamic human hair from multi-view videos. We propose an efficient and effective optimization method that is significantly faster compared to state-of-the-art dynamic human hair reconstruction methods, while achieving comparable reconstruction quality. The performance of our algorithm is demonstrated both qualitatively and quantitatively on the public NeRSemble dataset.

## 1 Introduction

Reconstructing dynamic human hair is a core task for acquiring high-quality and natural hair assets, which is crucial for modeling 3D talking heads or human avatars. Traditional methods model human hairs with geometric primitives Paris et al. (2004); Wei et al. (2005); Nam et al. (2019), which admit limited rendering quality. More recently, researchers propose using unstructured neural representations (e.g., NeRF and gaussian splattings), which ignores the inherent strand structure of human hair, making it difficult to animate the reconstructed human hair. In light of this, multiple research works Luo et al. (2024); Zakharov et al. (2024) propose using strand gaussians to represent human hair, which achieves superior rendering quality while maintaining the strand structure.

However, dynamic human hair reconstruction typically involves optimization procedures that work on high-resolution videos from tens or hundreds of cameras, which can take hours to days to process on a single GPU. In this work, we propose a series of acceleration techniques that enjoy orders of magnitude speed-up without sacrificing reconstruction quality.

Gaussian-based representation typically requires a large number of gaussians to accurately capture the geometry and dynamics of objects. The complicated dynamic densification strategy makes it

difficult to design acceleration strategy. In addition, strand gaussian representation imposes more constraints between gaussians, adding an extra layer of complexity.

In this work, we propose using the sparsity-driven approach to accelerate the reconstruction of dynamic human hair. In short, we make several adjustment to the previous strand gaussian representation to promote decoupling, so that a large fraction of non-active variables can be faithfully detected and isolated for a large fraction of the optimization procedure.

Looking at the rendering equation Kerbl et al. (2023) of a specific camera pixel $p$

$$\boldsymbol{C} = \sum_i T_i \alpha_i \boldsymbol{c_i}, \quad T_i = \prod_{j=1}^{i-1} (1 - \alpha_j) \quad (1)$$

where $\alpha_i$ and $\boldsymbol{c_i}$ is the final opacity and color of Gaussian. The accumulated transmittance term $T_i$ requires a sorting of all gaussians that intersect with the camera ray through $p$, which makes decoupling individual Gaussians difficult. Inspired by the sort-free representation proposed by Hou et al. (2024), we propose to replace the accumulated transmittance term with pixel-wise weighting function that blends the color of gaussians according to their depth. An immediate advantage of this modification is that once the weighting function is fixed, gaussians can be optimized independently.

We also observe that previous strand gaussian-based approaches spend a large fraction of time on globally align the hair strands. Therefore, we apply a coarse-to-fine optimization strategy that gradually densify the hair strands, significantly accelerating the convergence of the global alignment phase. Bringing the optimization to a local adjustment regime.

In the local optimization regime, hair strand structure and pixel-wise weighting function admit less changes and can be fixed for most of the time, suggesting that individual gaussians are independent and can be treated as point gaussians. Hence, we design an active set method that maintains the active gaussians. By leveraging the inherent sparsity of active gaussians, we iteratively optimize a much smaller set of active gaussians while converging on the same number of iterations, leading to a significant acceleration.

To sum up, our contributions are:

- We propose a novel dynamic human hair reconstruction methods via strand gaussians, which to our knowledge is the first such method in the community.

- We propose multiple modifications to different stages of the optimization procedure, aiming to promote sparsity-driven efficient optimization techniques.

- Our acceleration techniques greatly improve the computation efficiency without significant loss of reconstruction quality. We achieved approximately a 5x speedup in static scenes and about a 1.5x speedup in dynamic scenes compared to our non-optimized version.

## 2 RELATED WORKS

### 2.1 STATIC HAIR RECONSTRUCTION

Hair modeling has been recognized as a challenge in human avatar reconstruction, with representations including geometric primitives, neural/volumetric representation. For geometric primitives, Paris et al. (2004); Wei et al. (2005); Nam et al. (2019) utilize a sequence of line segments to represent hair. Additionally, Takimoto et al. (2024) uses guiding hair and differentiable rendering to reconstruct internal hair flow. However, hair reconstruction with geometric representations remains challenging when dealing with various hairstyles.

Some researches have employ neural representation to reconstruct hair. Kuang et al. (2022); Wu et al. (2022) utilizes a neural implicit representation to query the hair geometry and generate hair strands. In contrast, volumetric/point representation provides a way for hair modeling. Rosu et al. (2022) compresses the geometric and appearance information of hair into scalp texture and decoded scalp texture into explicit hair representation. Furthermore, Sklyarova et al. (2023) employs both volumetric and strand-based representations to reconstruct hair. The 3DGS Kerbl et al. (2023) introduces Gaussian volumes, providing an explicit representation for hair modeling. Zakharov et al.

(2024) decodes the implicit hair texture representation into 3D Gaussians. Luo et al. (2024) represents hair as a sequence of cylindrical 3D Gaussian primitives explicitly. However, the complexity of hair motion makes it challenging to directly transfer static models to dynamic scene.

## 2.2 DYNAMIC HAIR RECONSTRUCTION

Dynamic hair reconstruction compresses hair deformation from videos. Xu et al. (2014) combines optical flow and the orientation map to represent the motion of the hair. Liang et al. (2018) retrieves and matches hair deformations from a hair database based on morphable model (3DMM) Blanz & Vetter (1999). Similarly, Winberg et al. (2022) utilize head mesh deformations to estimate the rigid transformation of hair and further refine it. Wang et al. (2022) utilize a hybrid neural volumetric representation to modeling hair motion from the optical flow. Wang et al. (2023) design a variational auto-encoder (VAE) network to predict the implicit hair representation in time sequences. Explicit 3D Gaussian representations allow neural networks to directly predict the deformation compare with the implicit representations. Wang et al. (2024a) employes 3D Gaussians to represent the hair and use a multilayer perceptron (MLP) network to predict non-rigid deformations of hair motion. Qian et al. (2024) binds the hair representation in flame mesh. However, the representation using only 3D Gaussians is insufficient to accurately model the motion of complex hairstyles. Therefore, it is essential to model hair deformation with explicit 3D Gaussians representation.

## 2.3 ACCELERATION

Recent researches have demonstrate that the 3DGS pipeline can be significantly accelerated. Some works focus on reducing the number of Gaussians Rota Bulò et al. (2025); Fang & Wang (2024); Kheradmand et al. (2024); Lee et al. (2024); Hanson et al. (2024). C3DGS Lee et al. (2024) introduces a learnable mask to eliminate Gaussians that contribute minimally to the final rendering quality, Speedy-Splat Hanson et al. (2024) applies both soft and hard pruning techniques for accelerating. Other works Girish et al. (2025); Fan et al. (2023); Wang et al. (2024b); Mallick et al. (2024) aim to minimize computational overhead and memory consumption through Gaussian pruning, quantization, and efficient training strategies. EAGLES Girish et al. (2025) adopts quantized embeddings to reduce per-point memory usage and employs a coarse-to-fine training strategy. AdR-Gaussian Wang et al. (2024b) accelerates 3DGS by culling of Gaussian-tile pairs with low opacity. Additionally, Taming 3DGS Mallick et al. (2024) combines with per-splat parallelized backpropagation to enable efficient training. DashGaussian Chen et al. (2025) proposes a resolution scheduling method which fits 3DGS to higher level of frequency in the optimization phase. Above researches are mainly focus on point-based Gaussians, while there are hardly any research in strand Gaussians.

## 3 PRELIMINARY

In this section, we introduce the problem setting of dynamic hair reconstruction; the strand gaussian representation and its rendering equation; and the data preparation pipeline.

**Problem Setting.** Given multi-view videos $\mathcal{F} = \{F_1, ..., F_M\}$ from $M$ views, where each $F_i \in \mathbb{R}^{T \times H \times W \times C}$ is a stack of $T$ image frames from a single-view video, $H, W, C$ are height, width and RGB channels, we aim to reconstruct the geometry and appearance of human hair. Following the strand gaussian representation from Luo et al. (2024), we denote the geometry and appearance of each hair strand as $G_i$ and $A_i$, respectively. The geometry of a hair strand is represented by a series of gaussians $G_i = \{\boldsymbol{g}_{i,k} | k = 1, \ldots, K\}$, where $\boldsymbol{g}_{i,k} : \mathbb{R}^3 \to \mathbb{R}^+$ is a spatial density function:

$$\boldsymbol{g}_{i,k}(\boldsymbol{x}) = e^{-\frac{1}{2}(\boldsymbol{x}-\boldsymbol{\mu})^T \boldsymbol{\Sigma}^{-1}(\boldsymbol{x}-\boldsymbol{\mu})}, \tag{2}$$

where $\boldsymbol{\mu}$ and $\boldsymbol{\Sigma}$ are the center position and covariance matrix. The covariance matrix $\boldsymbol{\Sigma} = \boldsymbol{R}\boldsymbol{S}\boldsymbol{S}^T\boldsymbol{R}^T$ is derived from the rotation matrix $\boldsymbol{R}$ and diagonal scaling matrix $\boldsymbol{S} = \text{diag}(d, d, s)$, where $d = 10^{-4}$ represents the hair radius and $s$ represents the hair length. To satisfy the geometric constraint so that strand Gaussians are connected, the strand Gaussians satisfy the constraints:

$$\mu_i + \frac{1}{2}s_i\boldsymbol{d}_i = \mu_{i+1} - \frac{1}{2}s_{i+1}\boldsymbol{d}_{i+1}, \tag{3}$$

where $\boldsymbol{d}_i$ is the third column of $\boldsymbol{R}$, encoding the hair orientation direction. For convenience, we define the endpoints of gaussians $\boldsymbol{p}_{i+1} = \boldsymbol{\mu}_i + \frac{1}{2}s_i\boldsymbol{d}_i$. From the root of the hair on the scalp $\boldsymbol{p}_0$, we

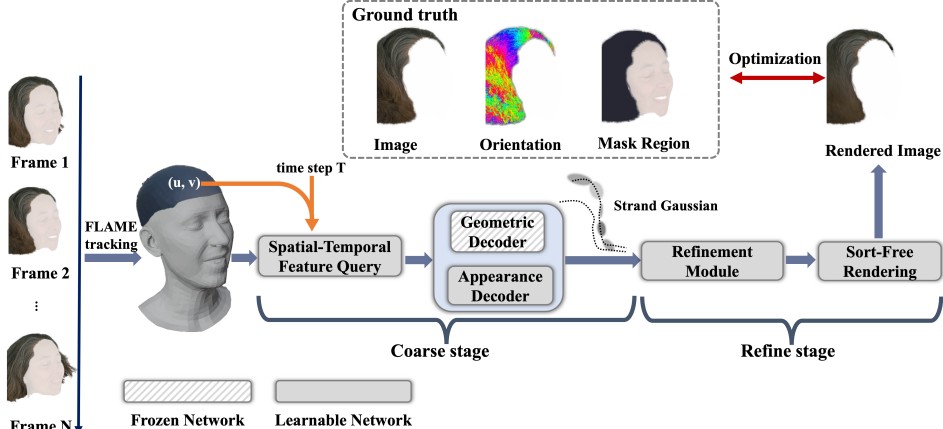

Figure 2: Our dynamic hair reconstruction algorithm takes multi-view videos and optimize for dynamic hair via strand gaussian representation. We bind the strand gaussians to scalp of FLAME face mesh. Our hair model consists of a strand gaussian generation module that generate a coarse set of guiding hair strands, followed by a refinement module that finetunes the interpolated hair strands for better rendering quality. In addition, we adopt the sort-free rendering module and other modifications to significantly accelerate the computation.

can accumulate each strand gaussian to generate the hair strand. The appearance of the hair strand is represented by a vector $A_i$ that consists of several attributes, including the opacity $\alpha_i \in \mathbb{R}^+$ and the RGB color of each view direction encoded with degree-3 spherical harmonics (SH) $h_i \in \mathbb{R}^{4^2 \times 3}$.

**Rendering Strand Gaussian.** Following the rendering equation of 3DGS Kerbl et al. (2023), we project the gaussian onto image space. The 2D covariance matrix can be derived by

$$\Sigma' = JW\Sigma W^T J^T, \tag{4}$$

where $J$ is the Jacobian matrix of the affine approximation of the projective transformation, and $W$ is the view transformation matrix. The rendering equation is similar as 1, where $c_i \in \mathbb{R}^3$ is the learned color. $\alpha_i'$ is the final opacity which denoted as:

$$\alpha_i' = \alpha_i \cdot \left( -\frac{1}{2} \left( x_i' - \mu_i' \right)^T \Sigma_i'^{-1} \left( x_i' - \mu_i' \right) \right), \tag{5}$$

where $x_i'$ and $\mu_i'$ are projected and center position in 2D space, and $\alpha_i$ is the opacity.

**Hair Perception.** Dynamic hair reconstruction requires additional perception information to ensure an accurate reconstruction of strand orientations. Therefore, we adopt perception model Yao et al. (2024) to segment the hair regions for each video frame, resulting in a binary mask $M_h \in [0, 1]^{H \times W}$. We also adopt the Gabor filter method from Zakharov et al. (2024) to compute pixel-wise hair orientation and confidence, denoted as $O_d \in \mathbb{R}^{H \times W \times 2}$ and $O_v \in \mathbb{R}^{H \times W}$, respectively.

**Dynamic Head Reconstruction.** To disentangle the head motion and hair motion, we adopt FLAME Li et al. (2017) to track the head poses over each time steps, resulting in a template mesh $M_t$ with predefined scalp region at each time step. During the reconstruction procedure, we fix the head pose and grow hair strands from root points on the scalp to ensure an accurate hair reconstruction.

## 4 APPROACH

In this section, we first introduce in Section 4.1 a novel dynamic hair reconstruction algorithm via strand Gaussians, which is adapted from state-of-the-art static hair reconstruction methods. We then describe in Section 4.2 and Section 4.3 the optimization procedure and our acceleration method that significantly improves the efficiency of the optimization procedure, respectively.

## 4.1 DYNAMIC HAIR RECONSTRUCTION VIA STRAND GAUSSIANS

Inspired by the graphics industry to reduce the number of strands, we explicitly model a set of coarse-grained optimizeable guiding hairs and densify them via interpolation. This strategy imposes smoothness prior and greatly improves computation efficiency. Our hair model consists of two components: 1) the strand generation module and 2) the refinement module. We will elaborate each of the modules and describe the optimization procedure of dynamic hair reconstruction.

**Strand Generation Module.** To represent strand-wise geometry, appearance, and dynamics, we work on a spatial-temporal space that combines the discretized scalp UV space and T time steps with a total dimensionality $|U| \times |V| \times T$. To avoid optimizing a 4D tensor as large as $|U| \times |V| \times T \times D$, we use the $K$-Plane Fridovich-Keil et al. (2023) technique that combines pairwise position-encoded features into strand-wise feature vectors $C(u, v, t) \in \mathbb{R}^{D_C}$. Specifically, we have

$$C(u,v,t) = C_{uv}(u,v) \circ C_{vt}(v,t) \circ C_{ut}(u,t) \qquad (6)$$

where $\circ$ is element-wise multiplication, $C_{uv} \in \mathbb{R}^{|U| \times |V| \times D_C}$, $C_{vt} \in \mathbb{R}^{|V| \times T \times D_C}$ and $C_{ut} \in \mathbb{R}^{|U| \times T \times D_C}$.

While previous works such as Zakharov et al. (2024) and Rosu et al. (2022) propose to process $C$ via a UNet architecture. We argue that such approaches unnecessarily entangle variables, making it difficult to employ sparsity-driven optimization techniques. Instead, we propose to independently process each strand-wise feature $C(u, v, t)$ using a MLP, resulting in strand-wise feature $L_i \in \mathbb{R}^{D_L}$.

We adopt the strand geometry decoder and its pretrained weights from Zakharov et al. (2024) to decode the strand geometry $G_i$. Specifically, the geometry decoding network takes the first $\frac{D_L}{2}$ channels from the strand-wise feature $L_i$ and output the displacements $\{\boldsymbol{d}_j | j = 1, \ldots, K\}$ of each strand defined in equation 3. During optimization, the strand geometry decoder is frozen to implicitly impose a data-driven strand geometry prior and improve the optimization stability.

We also design a similar network for decoding strand appearance, which takes the last $\frac{D_L}{2}$ channels from $L_i$ and outputs the spherical harmonic coefficients $\{\boldsymbol{h}_j \in \mathbb{R}^{48} | j = 1, \ldots, K\}$ and the confidence $\{\gamma_j \in \mathbb{R}^+ | j = 1, \ldots, K\}$ of the detected orientation from data preprocessing.

To improve the smoothness of hair strands and reduce computational complexity, we densify the guiding hair strands by bilinear interpolation on the scalp UV space.

**Refinement Module.** While strand-based gaussians impose strong structural priors of human hairs, they typically admit a gap in the reconstruction quality due to over-smoothing on the strand geometry and the guiding hair strand interpolation mechanism. To mitigate this issue, we propose to add an additional refinement module to the strand gaussians that allows each interpolated strand to slightly vary in geometry and appearance. Specifically, we introduce a set of delta variables on the strand position $\Delta v_{i,j}$ and on the spherical harmonics coefficient vector $\Delta h_i$ for all interpolated hair strands.

## 4.2 OPTIMIZATION

The optimization procedure of our dynamic hair reconstruction algorithm is divided into the coarse and the refine stage. We adopt the traditional 3DGS Kerbl et al. (2023) objectives $\mathcal{L}_1$ and $\mathcal{L}_{\text{SSIM}}$, which focuses on the rendering difference. To enforce the hair strand structure and appearance, we introduce the silhouette loss $\mathcal{L}_{mask}$ Zakharov et al. (2024) to constraint that the 2D projection of hair strands matches the predicted hair segmentation mask. We also use the hair orientation direction detected on images (see Hair Perception in Section 3) to regularize the 3D strand orientation direction $d_i$. To this end, we project each 3D orientation direction to pixel $P$ and define the orientation loss

$$\mathcal{L}_{ori} = \sum_P \left\| \sum_{i=1}^{N_P} T_i d_i^P \alpha_i' - O_d(P) \right\|, \qquad T_i = \prod_{j=1}^{i-1} \left(1 - \alpha_j'\right), (7)$$

where $d_i^P$ is the projected 2D orientation, similar to equation 1.

**Coarse Optimization Stage.** During the coarse stage, we disable the refinement module and focus on optimizing the strand generation module. The coarse optimization stage ends when the descent amount of the objective function at a certain resolution is smaller than a predefined threshold $\epsilon$.

To improve stability of the optimization procedure, we add a smooth regularization term $L_{\text{smooth}}$ as

$$\mathcal{L}_{\text{smooth}} = \sum_{i=0}^{N} \sum_{j=1}^{L-1} \|\boldsymbol{v}_{i,j+1} - \boldsymbol{v}_{i,j}\|, \quad \boldsymbol{v}_{i,j+1} = \boldsymbol{p}_{i,j+1} - \boldsymbol{p}_{i,j} \tag{8}$$

The optimization objective function in coarse stage is

$$\mathcal{L}_{\text{coarse}} = \mathcal{L}_1 + \mathcal{L}_{\text{SSIM}} + \mathcal{L}_{mask} + \mathcal{L}_{ori} + \mathcal{L}_{\text{smooth}} \tag{9}$$

**Refine Optimization Stage.** Strand gaussians from the converging state of coarse optimization exhibits reasonably good strand geometry and position, yet there is still a gap on the rendering quality. The refine optimization stage freezes the strand gaussian generation module and finetunes the strand geometry and appearance by introducing the delta strand positions $\Delta v_{i,j}$ and the delta appearance represented by spherical harmonics coefficients $\Delta h_j$. We impose regularization terms on the norm the delta variables and end up with an optimization objective for refine stage as

$$\mathcal{L}_{\text{refine}} = \mathcal{L}_1 + \mathcal{L}_{SSIM} + \mathcal{L}_{mask} + \mathcal{L}_{ori} + \lambda_v \|\Delta v\|^2 + \lambda_h \|\Delta h\|^2 \tag{10}$$

### 4.3 ACCELERATION FOR DYNAMIC HAIR RECONSTRUCTION

We propose three modifications to the dynamic hair reconstruction algorithm introduced in Section 4.1 and 4.2, aiming to employ sparsity-driven efficient optimization techniques from traditional machine learning optimization theory. The major motivation of all three modifications is to make gaussian variables decoupled from each other, so that sparsity-driven approaches can freeze the non-active gaussian variables and focus on a much smaller number of active ones.

**Sort-Free Rendering.** First, by looking at the rendering equation 1. We observe that for non-transparent objects, the accumulated transmittance term $T$ takes exponentially small value for the occluded gaussians, leading to unnecessary coupling of variables. Inspired by the Sort-Free Gaussians Splatting Hou et al. (2024), we design a weighting module that takes a camera ray direction $r$ and a depth value $d$ of a specific gaussian and outputs the weight $w(r, d)$ to replace the accumulated transmittance term $T$, leading to a new rendering equation $C = \frac{\sum_i w(r_i, d_i) c_i}{\sum_i w(r_i, d_i)}$. When the weighting module is fixed, the optimization of $c_i$ can be decoupled.

**Coarse-to-Fine Optimization.** During the coarse optimization stage, it is difficult to decouple gaussian variables as they are related by the strand structure and the motion. Therefore, we adapt the classic coarse-to-fine optimization technique to accelerate this stage. We start with 256 guiding hair strands on a discretized scalp UV space with $|U| = |V| = 16$. Once the optimization converges with an objective value larger than $\epsilon$, we double $|U|$ and $|V|$ and restart the coarse optimization. The coarse-to-fine strategy leads to roughly 1.5x speed-up in the coarse stage.

**Active Set Optimization.** With the previous modifications, we employ the active set strategy in the late phase of the coarse stage and the entire refine stage by detecting active strands and gaussians, respectively. For non-active strands and gaussians. We use pre-render technique to store the rendering images of all non-active variables. Thanks to the Sort-Free rendering, the final rendering image is simply the addition of pre-rendered image and the rendering result of active strands and gaussians. In the coarse stage, the active set strategy leads to 1.2x-1.5x speed-up. In the refine stage, the active set strategy leads to 3x-4x speed-up in the static scene and 1.6x speed-up in the dynamic scene.

## 5 EXPERIMENTS

We presents an experimental evaluation of proposed approach, including comparable evaluation by novel view synthesis and the ablation study.

### 5.1 EXPERIMENTAL SETUP

**Dataset.** We use the NeRSemble Kirschstein et al. (2023) dataset for training and evaluation. The recordings in NeRSemble are captured simultaneously from multiple viewpoints and include movements of various parts of face. For each subject, we select 16 viewpoints that cover the hair regions and downsample the images to a resolution of $550 \times 802$, which is one-quarter of the original size.

**Baselines.** We consider two categories of baselines for static and dynamic hair reconstruction. For static hair reconstruction, we quantitatively evaluate our method with point-based Gaussian methods (3DGS Kerbl et al. (2023) and AdR-Gaussian Wang et al. (2024b)) and a strand Gaussian method (Gaussian-Haircut Zakharov et al. (2024)). For dynamic hair reconstruction, we evaluate our method with four baselines: PointAvatar Zheng et al. (2023), 4DGS Wu et al. (2024), GaussianAvatars Qian et al. (2024), and Gaussian-Head-Avatars Xu et al. (2024). PointAvatar adopts a differentiable rendering with a deformable point-based avatar representation. The last three are point-based Gaussian methods. 4DGS optimizes point-based Gaussians in dynamic scenes, while GaussianAvatars and Gaussian-Head-Avatars using point-based Gaussians with dynamic head representations.

**Metrics.** We adopt the evaluation metrics used in Kerbl et al. (2023), including LPIPS, SSIM, and PSNR computed within the hair region, to assess rendering quality. The calculation of our region-specific PSNR is provided in equation 11. Notably, we exclude the facial region when computing metrics within the hair area to more accurately evaluate the quality of hair reconstruction.

$$\text{PSNR} = 10 \log_{10} \left( \frac{\text{MAX}_I^2}{\text{MSE}} \right), \quad \text{MSE} = \frac{1}{N} \sum_{(i,j) \in M} (R_{i,j} - GT_{i,j})^2 \tag{11}$$

where $R$ and $GT$ are rendered and ground truth image, $M$ is the assemble that pixel coordinates in hair region, containing $N$ pixels. We record the average training time to assess the efficiency.

**Computational Resources.** All experiments are executed on a single NVIDIA H800 GPU with 80 GB GPU memory.

**Implementation Details.** The detailed implementation settings are described in the Appendix A.

## 5.2 EVALUATION & COMPARISON

Our comparisons consist of two parts: Static hair reconstruction and Dynamic hair reconstruction.

**Static Hair Reconstruction**. We utilize the images from 15 viewpoints for training, and randomly select one of the remaining viewpoints as the test view. This random selection is repeated three times, and the final performance is reported as the average over the three runs. We compared our method with existing baselines: 3DGS Kerbl et al. (2023), AdR-Gaussian Wang et al. (2024b), and Gaussian-Haircut Zakharov et al. (2024). The results of subject 304 in NeRSemble are presented in Figure3 and Table1. In terms of appearance, point-based Gaussian methods exhibit superior performance, while Gaussian-Haircut Zakharov et al. (2024) and our method achieve visually comparable results. However, due to imprecise hair geometry, point-based Gaussian methods tend to produce locally blurry regions, as illustrated in Figure 3.

Table 1: Results of static hair reconstruction quantitative comparison

| Category | Method | PSNR(NVS) ↑ | SSIM(NVS)↑ | LPIPS(NVS) ↓ | runtime(s) ↓ |
|---|---|---|---|---|---|
| Point | 3DGS Kerbl et al. (2023) | **24.96** | **0.8955** | 0.0823 | 259.0 |
| | AdR-Gaussian Wang et al. (2024b) | 24.88 | 0.8949 | **0.0771** | **212.3** |
| Strand | Gaussian-Haircut Zakharov et al. (2024) | 23.70 | 0.8746 | 0.0860 | 7184 |
| | Ours | 24.52 | 0.8472 | 0.0876 | 1793 |

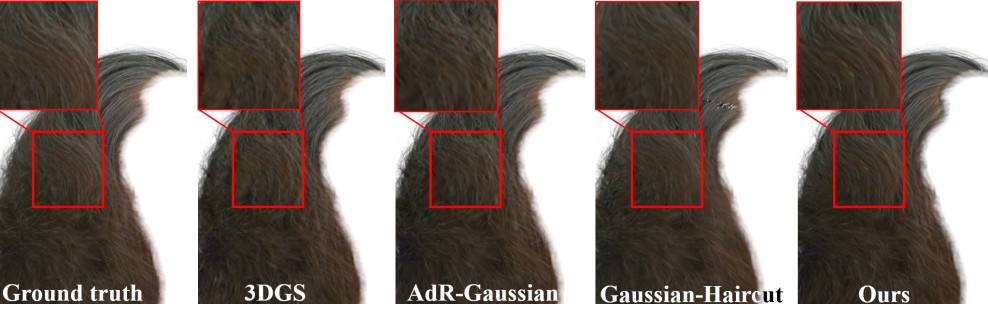

Figure 3: The comparing results of hair rendering on novel view for static methods. From left to right represents the ground truth, the results of 3DGS, AdR-Gaussian, Gaussian-Haircut, our method

For efficiency evaluation, we measure the average training runtime. Benefiting from the simplicity of the point-based representation, 3DGS Kerbl et al. (2023) and AdR-Gaussian Wang et al. (2024b) demonstrate better training efficiency. Nonetheless, the strand Gaussian representation excels in dynamic hair reconstruction, as it better incorporates structural constraints of hair. While our method is slower than point-based Gaussian approaches, it achieves improved training efficiency compared to Gaussian-Haircut and delivers a fourfold acceleration, as reported in Table 1.

**Dynamic Hair Reconstruction.** We conduct experiments with subject 304 in the NeRSemble dataset Kirschstein et al. (2023), comparing with four baselines: Point-Avatar Zheng et al. (2023), 4DGS Wu et al. (2024), Gaussian-Avatar Qian et al. (2024), and Gaussian-Head-Avatar Xu et al. (2024)(Note: for efficiency, we truncate the iterations times of Gaussian-Head-Avatar to limit training time). Similar to static hair reconstruction, we evaluate using both appearance and efficiency metrics. Due to its point-based representation and fixed point size, Point-Avatar yields the worst performance for appearance metrics. Additionally, 4DGS demonstrates inferior appearance quality compared to Gaussian-Avatar and Gaussian-Head-Avatar, primarily because it lacks the structural prior provided by the FLAME mesh. The FLAME mesh offers an effective initialization for dynamic modeling, requiring only the prediction of 3D Gaussian deformations relative to the mesh. Consequently, both Gaussian-Avatar and Gaussian-Head-Avatar achieve better appearance quality than 4DGS. However, they do not consistently outperform 4DGS in Figure 4, as hair in these methods is constrained to the underlying mesh, limiting their ability to represent dynamic hair motion accurately. In contrast, our strand Gaussian, which grows from the scalp mesh, enables more flexible and accurate modeling of hair deformations. As shown in Table 2 and Figure 4, our method achieves the best appearance results for dynamic hair reconstruction. Due to page limitations, the hair reconstruction results of other subjects are provided in the Appendix A.

Table 2: Results of dynamic hair reconstruction quantitative comparison

| Category | Method | PSNR(NVS)↑ | SSIM(NVS)↑ | LPIPS(NVS)↓ | runtime(s)↓ |
|---|---|---|---|---|---|
| Point | Point-Avatar Zheng et al. (2023) | 17.65 | 0.8102 | 0.1687 | 4825 |
| | 4DGS Wu et al. (2024) | 18.36 | **0.8595** | 0.1519 | **3727** |
| | Gaussian-Avatar Qian et al. (2024) | 21.14 | 0.8881 | 0.1018 | 12836 |
| | Gaussian-Head-Avatar Xu et al. (2024) | 21.76 | 0.8753 | 0.1271 | 11620 |
| Strand | Ours | **24.28** | 0.8790 | **0.0964** | 4397 |

Ground truth    Gaussian-Avatar    Point-Avatar    4DGS    Gaussian-Head-Avatar    Ours

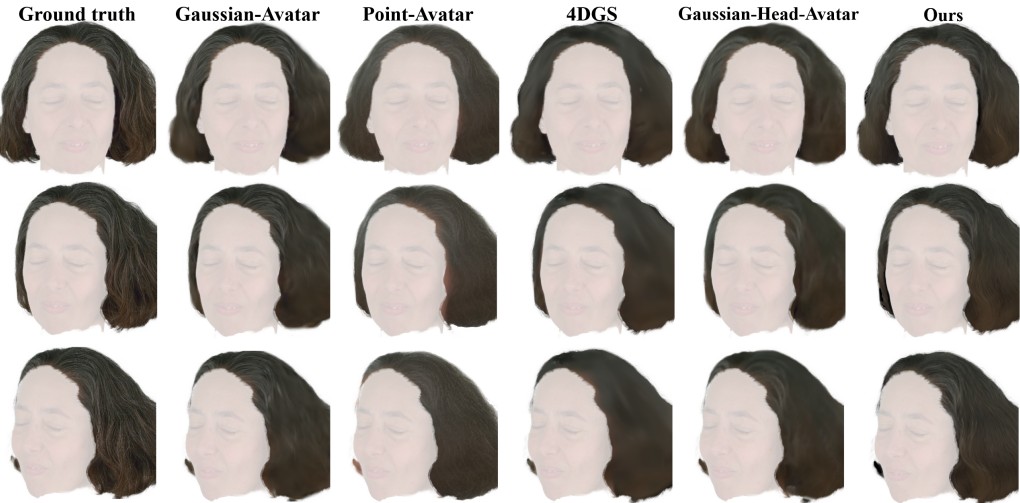

Figure 4: The comparing results of hair rendering on novel view for dynamic methods. From left to right represents the ground truth, the results of Gaussian-Avatar, Point-Avatar, 4DGS, Gaussian-Head-Avatar, and our method

We incorporate a sparsity-driven module, which yields faster performance than Point-Avatar, Gaussian-Avatar, and Gaussian-Head-Avatar, with only a marginal slowdown compared to 4DGS. However, it is challenging to determine whether the observed appearance quality arises from the method itself or from extended training duration. We present the PSNR-over-time curve with one viewpoint in Figure 5, where the y-axis represents PSNR and the x-axis denotes time. As shown in Figure 5, our method achieves the best performance among the four baselines.

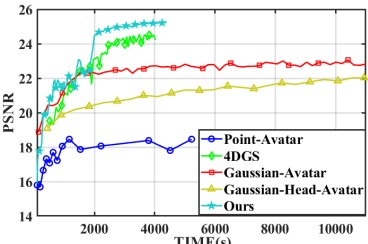

Figure 5: Dynamic hair reconstruction quantitative comparison on PSNR and training time

## 5.3 ABLATION STUDY

We evaluate three components of our method to validate their efficiency and effectiveness.

$K$**-Plane Module in UV Grid Space.** We replace the $K$-Plane module with position-encoded coordinates in UV space across $T$ time steps to evaluate our method. The evaluation results for different time steps are shown in the left sub-table of Table 3. The appearance quality achieved by our method with the $K$-Plane module is clearly superior, as the position-encoded coordinates in UV space and $T$ time steps fail to capture temporal information across the sequence.

**Refinement Module.** We evaluate the effectiveness of the refinement module. We remove the refinement module and measure the resulting appearance quality. The evaluation results are presented in the right sub-table of Table 3. As shown, the appearance quality is significantly improved when the refinement module, while it enables more accurate modeling of the temporal deformation of hair.

**Sparsity-driven Acceleration.** We adopt a coarse-to-fine strategy combined with scalp-grid sparsity, while keeping the maximum grid resolution unchanged in coarse stage. In refine stage, we employ a pre-rendering technique to accelerate the rendering process. As the number of training images is too large to pre-render all frames in dynamic scene, we render Gaussians only in the active set. For the inactive Gaussians, we fix the gradients during backpropagation to further reduce computational overhead. As illustrated in in Figure 6, our sparsity-driven design leads to approximately an 50% improvement in training efficiency for static scene and 60% for dynamic scene.

Table 3: Ablation study results on $K$-Plane module in grid space (left) and refinement module (right)

| Grid | PSNR(NVS)↑ | SSIM(NVS)↑ | LPIPS(NVS)↓ | Refinement | PSNR(NVS)↑ | SSIM(NVS)↑ | LPIPS(NVS)↓ |
|---|---|---|---|---|---|---|---|
| ✗ | 20.40 | 0.8705 | 0.1077 | ✗ | 24.94 | 0.8998 | 0.0779 |
| ✓ | 23.87 (+3.47) | 0.8933 (+0.0228) | 0.0845 (-0.0232) | ✓ | 25.27(+0.33) | 0.9037(+0.0039) | 0.0737(-0.0042) |

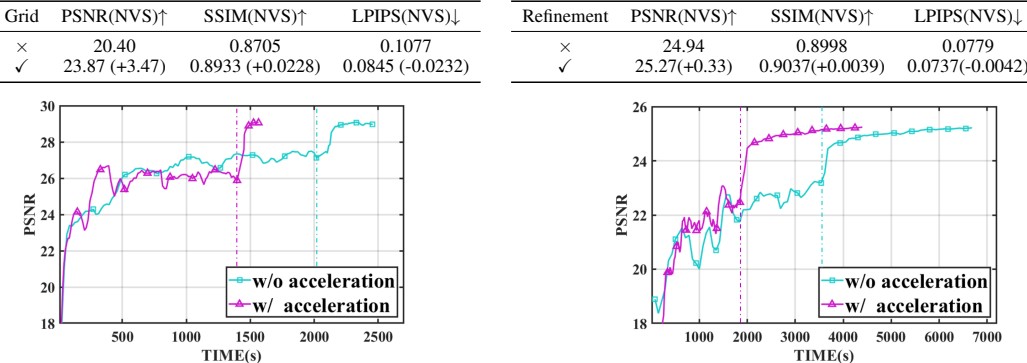

Figure 6: The results of ablation study on sparsity-driven acceleration. The left shows the static reconstruction while the right shows the dynamic reconstruction

## 6 CONCLUSION AND LIMITATIONS

**Conclusion.** In this work, we present the first to our knowledge dynamic hair reconstruction method via strand gaussian representation. In addition, we made multiple modifications to promote the sparsity-driven approaches, which significantly improves the computation efficiency.

**Limitations.** The current acceleration techniques have limited effect for the coarse stage optimization since the initialization for hair strand geometry are poor. We thus believe that by improving the hair perception model or employing large reconstruction model for human hairs, the total computation time of the coarse optimization stage can be greatly reduced, leading to much better relative improvement on the computation time for our proposed acceleration techniques.

## 7 ETHICS STATEMENT

This research does not involve studies with human subjects, nor does it raise concerns related to data collection from individuals. No practices associated with dataset release are included, and the work does not generate or disseminate potentially harmful insights, methodologies, or applications.

There are no conflicts of interest, external sponsorships, or financial influences that could affect the objectivity of this work. Issues of discrimination, bias, and fairness are not implicated in the study, and no privacy or security risks are introduced.

The research fully adheres to applicable legal and regulatory requirements, and it has been conducted in accordance with principles of academic honesty and integrity.

## 8 REPRODUCIBILITY STATEMENT

We affirm that the experiments reported in this work are reproducible. To facilitate verification and replication of our results, we have provided the implementation details in Appendix A and source code in the supplementary materials. The included code allows reviewers to reproduce the main experiments and confirm the conclusions presented in this manuscript.

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

## A  APPENDIX

**Brief Introduction.**The appendix is structured into three main sections: experiment settings, supplementary experiments, and experimental statistical analysis. The main contents are as follows:

**Experiment Settings**. We use Adam Kingma & Ba (2017) for parameter optimization. We set the learning rate to $10^{-3}$ for the spatial-temporal feature query network and appearance decoder in coarse stage, $1.6 \times 10^{-6}$ for the position and keep the same learning rates as 3DGS Kerbl et al. (2023) in refine stage. We set the loss weight 0.1 to orientation, 0.1 to mask, and 0.05 for hair smoothness. We train the 20000 iterations for coarse stage and 10000 iterations for refine stage. In the first 10000 iterations, we conduct coarse-to-fine method and last 10000 for active set method. Additionally, we conduct the active method in the whole refine stage. In our method, the number of hair strands is set to 20,000, with each strand consisting of 100 nodes, corresponding to 99 Strand Gaussians per strand. The root nodes of our hair strands are generated from the scalp, which is derived from fixed index positions on the FLAME Li et al. (2017) mesh.

**Supplementary Experiments**. We evaluate both static and dynamic hair reconstruction using two additional subjects from the NeRSemble dataset.

In static hair reconstruction, the results of subject 199 in NeRSemble are shown in Figure 7, while Figure 8 presents the results for subject 214. The enlarged local regions of the images demonstrate that strand Gaussians provide a more effective representation of hair appearance.

In dynamic hair reconstruction, the results for one participant are shown in Figure 9, whereas Figure 10 presents the results for the other. The results of dynamic hair reconstruction demonstrate that strand Gaussians effectively maintain structural consistency throughout motion, leading to improved performance in both geometric accuracy and appearance representation.

In the previous section, we present experimental results on subject 304, 199 and 214. Additionally, we provide quantitative appearance experimental results of dynamic hair reconstruction for three more subjects 104, 226 and 293 in Table 4 and three more viewpoints 222200038, 222200042 and 221501007 in Table 5. In the experiment, we compare our method with two top-performing baselines: Gaussian-Avatar and Gaussian-Head-Avatar.

Table 4: Results of dynamic hair reconstruction quantitative comparison with more subjects

| Subject | Method | PSNR(NVS)↑ | SSIM(NVS)↑ | LPIPS(NVS)↓ |
|---------|--------|------------|------------|-------------|
| 104 | Gaussian-Avatar Qian et al. (2024) | 22.65 | **0.9501** | 0.0903 |
| 104 | Gaussian-Head-Avatar Xu et al. (2024) | 22.19 | 0.9388 | 0.1154 |
| 104 | Ours | **23.88** | 0.9025 | **0.0893** |
| 226 | Gaussian-Avatar Qian et al. (2024) | 20.29 | **0.9114** | 0.0909 |
| 226 | Gaussian-Head-Avatar Xu et al. (2024) | 20.59 | 0.9055 | 0.1254 |
| 226 | Ours | **21.62** | 0.9063 | **0.0774** |
| 293 | Gaussian-Avatar Qian et al. (2024) | 25.61 | 0.9659 | 0.0376 |
| 293 | Gaussian-Head-Avatar Xu et al. (2024) | 26.16 | 0.9629 | 0.0555 |
| 293 | Ours | **26.79** | **0.9691** | **0.0321** |

## B  STATEMENT ON THE USE OF LARGE LANGUAGE MODELS

In the preparation of this manuscript, We employed a Large Language Model (LLM) to assist with language refinement. The model was used exclusively for purposes of improving grammar, clarity, and stylistic coherence. No substantive modifications were made to the research content, data analysis, or conclusions.

All intellectual contributions, including the conceptual framework, methodology, data collection, analysis, and interpretation of results, are solely our own responsibility. The use of the LLM does not diminish the originality, independence, or academic integrity of this work.

Table 5: Results of dynamic hair reconstruction quantitative comparison with more viewpoints for subject 304

| viewpoints | Method | PSNR(NVS)↑ | SSIM(NVS)↑ | LPIPS(NVS)↓ |
|---|---|---|---|---|
| 222200038 | Gaussian-Avatar Qian et al. (2024) | 23.55 | 0.9236 | 0.0947 |
| 222200038 | Gaussian-Head-Avatar Xu et al. (2024) | 22.70 | 0.9165 | 0.1077 |
| 222200038 | Ours | **24.19** | **0.9282** | **0.0715** |
| 222200042 | Gaussian-Avatar Qian et al. (2024) | 20.81 | 0.8719 | 0.1243 |
| 222200042 | Gaussian-Head-Avatar Xu et al. (2024) | 21.70 | 0.8721 | 0.1524 |
| 222200042 | Ours | **21.76** | **0.8777** | **0.1049** |
| 221501007 | Gaussian-Avatar Qian et al. (2024) | 20.27 | 0.8850 | 0.1560 |
| 221501007 | Gaussian-Head-Avatar Xu et al. (2024) | 22.13 | 0.8888 | 0.1264 |
| 221501007 | Ours | **22.81** | **0.8906** | **0.1112** |

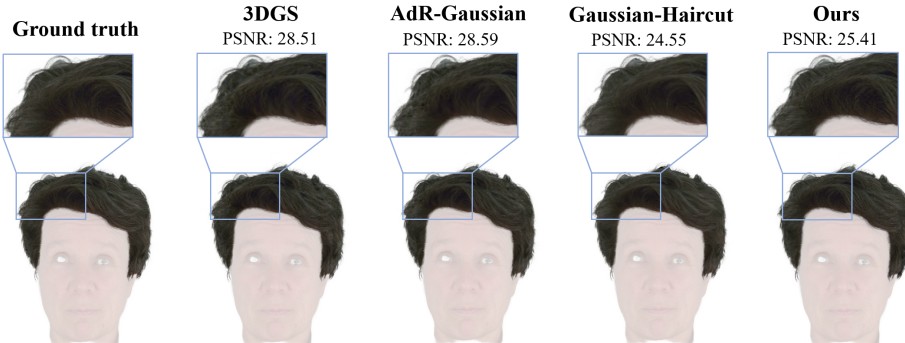

Figure 7: The comparing results of static methods for subject 199. From left to right represents the ground truth, the results of 3DGS, AdR-Gaussian, Gaussian-Haircut, our method. For better displayment, we increase brightness for partial enlarged images

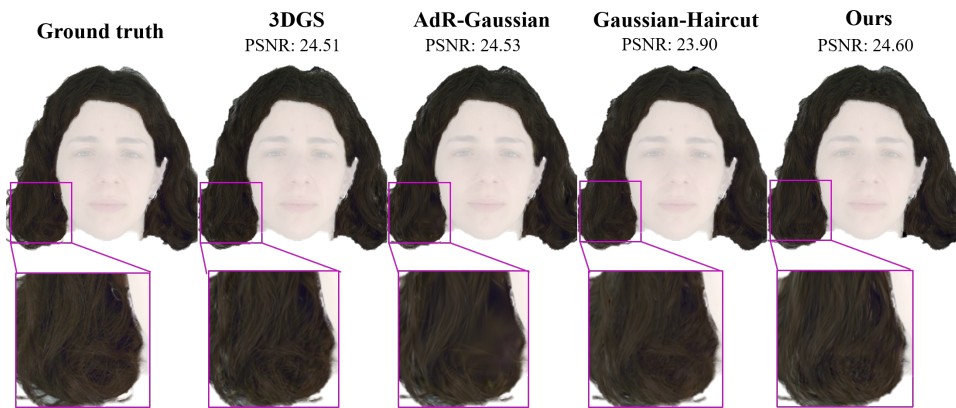

Figure 8: The comparing results of static methods for subject 214. From left to right represents the ground truth, the results of 3DGS, AdR-Gaussian, Gaussian-Haircut, our method. For better displayment, we increase brightness for partial enlarged images

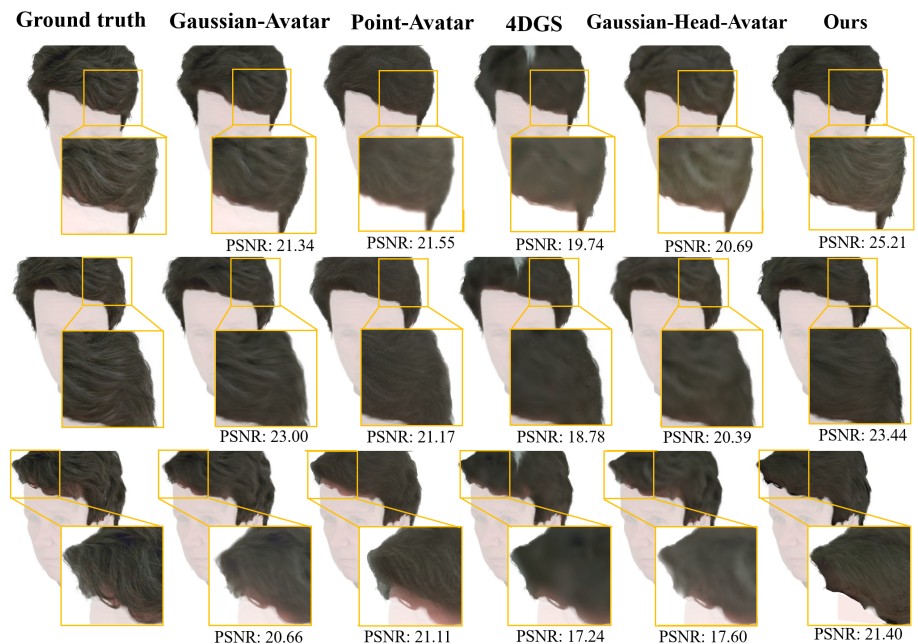

Figure 9: The comparing results of dynamic methods for subject 199. From left to right represents the ground truth, the results of Gaussian-Avatar, Point-Avatar, 4DGS, Gaussian-Head-Avatar, and our method. For better displayment, we increase brightness for partial enlarged images

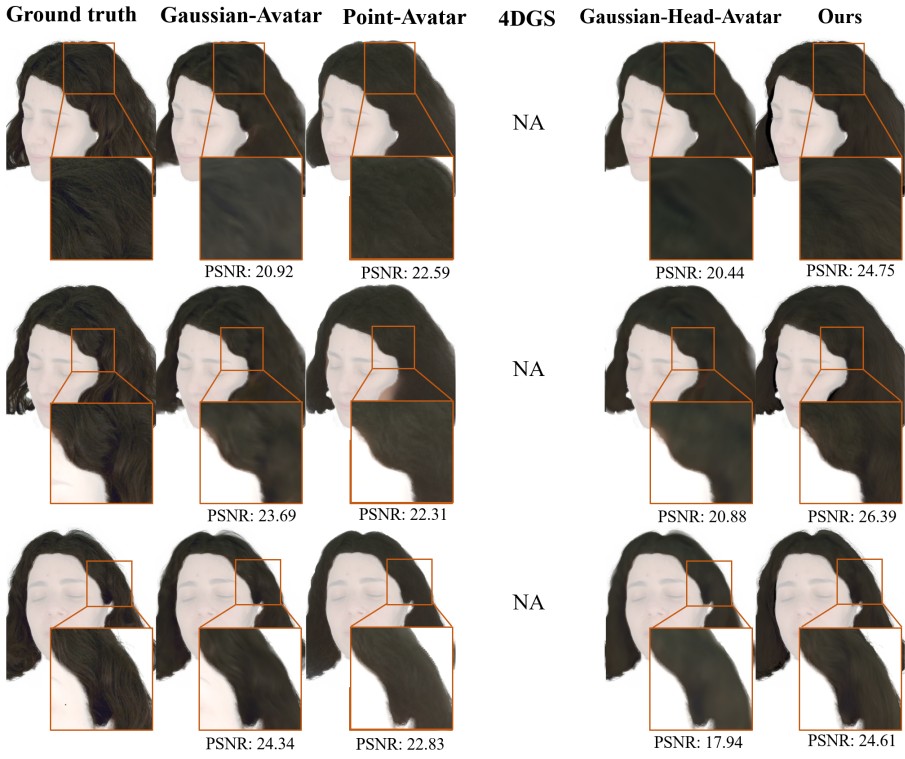

Figure 10: The comparing results of dynamic methods for subject 214. From left to right represents the ground truth, the results of Gaussian-Avatar, Point-Avatar, 4DGS, Gaussian-Head-Avatar, and our method. NA indicates that the methods (4DGS) failed in this reconstruction task. For better displayment, we increase brightness for partial enlarged images

