# OpenReview forum: "Efficient and Effective Dynamic Hair Reconstruction via Strand Gaussians"
_ICLR.cc/2026/Conference — ICLR 2026 Conference Withdrawn Submission_

### Official Review · Reviewer_K9rj · 2025-10-31

**Soundness:** 2
**Presentation:** 3
**Contribution:** 2
**Rating:** 2
**Confidence:** 3

**Summary:**

This paper proposes a pipeline for dynamic hair reconstruction using strand gaussians. The authors' biggest contribution is to improve training efficiency by optimizing the training process through methods such as sparsity-driven.

**Strengths:**

1. They seem to be the first to apply strand gaussians to dynamic hair reconstruction.

2. The entrie pipeline is reasonable, right methods are adapted where appropriate.

3. Based on their evaluation settings, the results are competitive compared to the chosen baselines.

**Weaknesses:**

1. The innovation is minimal. Basicly all parts of the methodology is directly derived from existing work without sufficient exploration or improvement. Especially in the dynamic parts, the authors seem to have made no contribution here, simply using methods from FLAME for motion modelling.

2. The accumulated transmittance actually contributes significantly to the final rendering quality of 3DGS. Methods that tried to prune low opacity gaussian or remove accumulated transmittance will mostly result in worse results. The Sort-Free Gaussian Splatting actually makes quite significant compromises on rendering quality (if you've tried with in-the-wild scenes). I don't think it's worth trading the rendering quality for limited efficiency improvement (from the table 2, maximum 3x faster, but still in hours).

3. Because sort-free Gaussian splatting degrades rendering quality, I was surprised to see the authors' method performed better than other baselines with dynamic scenes. Then I noticed that the chosen baselines are actually focused on the reconstruction of the entire head. However, the authors' comparison settings only considered the hair region. I can understand this is reasonable for a hair reconstruction work. But I still think it will be better to compare the entire head. The best outcome would be SOTA reconstruction results of other areas of the head, with a significant improvement in the hair region.

**Questions:**

Can the authors provide a evaluation result with other baselines for the entire head reconstruction?

---

### Official Review · Reviewer_rNiV · 2025-10-31

**Soundness:** 2
**Presentation:** 2
**Contribution:** 2
**Rating:** 2
**Confidence:** 4

**Summary:**

This paper proposes a dynamic human hair reconstruction method based on strand gaussians, aiming to reduce the computational cost of existing state-of-the-art methods while maintaining reconstruction quality. The method includes a strand generation module (using K-Plane for spatio-temporal feature fusion) and a refinement module (fine-tuning interpolated strands), along with three acceleration techniques: sort-free rendering, coarse-to-fine optimization, and active set optimization.

**Strengths:**

1.	The proposed acceleration techniques (sort-free rendering, coarse-to-fine optimization, active set optimization) target the computational bottlenecks of Gaussian-based hair reconstruction.
2.	The authors conduct experiments covering static and dynamic hair reconstruction, comparing against multiple baselines (point-based and strand-based). They also include ablation studies to verify the impact of key components (K-Plane module, refinement module, acceleration techniques) and supplementary experiments on additional subjects and viewpoints, which helps enhance the reliability of the results.

**Weaknesses:**

1.	The authors claim the method is the "first dynamic hair reconstruction method via strand gaussians," but do not fully explain how it differs from prior work that uses strand gaussians for static hair (e.g., Zakharov et al. 2024). The distinction between extending static strand Gaussians to dynamic scenes and creating a "first" dynamic method is not clearly articulated.
2.	While the authors note that acceleration is less effective in the coarse stage due to poor strand geometry initialization, they provide no specific plans or preliminary exploration to address this issue. There is also no detailed explanation of why coarse-stage sparsity is harder to exploit, which limits the understanding of the method’s limitations.
3.	For dynamic reconstruction, the authors truncate the iteration count of Gaussian-Head-Avatar to limit training time. This raises questions about whether Gaussian-Head-Avatar could achieve better quality with full training, making the comparison between the proposed method and this baseline less objective. The PSNR-over-time curve only covers one viewpoint, which is insufficient to fully address this bias.
4.	The Appendix mentions using 20,000 hair strands (with 99 gaussians each) but does not explain the rationale behind choosing this strand density.
5.	The title of this manuscript (i.e., “FORMATTING INSTRUCTIONS FOR ICLR 2026 CONFERENCE SUBMISSIONS”) is not right.

**Questions:**

Please refer to the weakness part.

---

### Official Review · Reviewer_1VQk · 2025-10-31

**Soundness:** 2
**Presentation:** 2
**Contribution:** 2
**Rating:** 4
**Confidence:** 3

**Summary:**

This paper presents a method based on Gaussian Splatting for dynamic hair reconstruction, aiming to improve the efficiency and quality. It first builds upon previous strand-based Gaussian splatting for hair representation, incorporating modifications such as coarse-to-fine optimization and densification to further improve training speed. The proposed method demonstrates superior performance compared to Gaussian-based avatar reconstruction methods on the NeRSemble dataset.

**Strengths:**

1. The paper is well-written and easy to follow, providing sufficient implementation details for reproduction.
2. The proposed techniques are reasonable and effectively contribute to improving speed and quality.

**Weaknesses:**

1. The paper's contribution appears limited. Primarily, the main contribution is extending strand Gaussian representation to dynamic scenes, which seems to be a straightforward combination. Additionally, the acceleration scheme is not technically significant; coarse-to-fine optimization for grid-based NeRF representations is a common practice in previous NeRF research, and the selection of rendering primitives has been well-studied in many compact 3D Gaussian splatting works [1]. Combining these improvements does not highlight a notable contribution. Furthermore, the proposed method only speeds up the training process by 1.5 times, which is not particularly impressive.

2. While the method outperforms baselines in dynamic hair reconstruction, these baselines were not specifically designed or optimized for this purpose. For example, the methods used do not incorporate strand orientations as this paper does, leading to an unfair comparison. Furthermore, when the proposed method is compared to strand-based Gaussian in static scenes, the improvement is minimal. The current manuscript does not clearly articulate the source of this gain.

[1] Mini-Splatting: Representing Scenes with a Constrained Number of Gaussians.

**Questions:**

The paper needs stronger baselines to highlight the significance of the proposed method. These baselines should primarily consider methods that extend static hair reconstruction with dynamic components for modeling, rather than methods for full-head avatars.

---

### Official Review · Reviewer_T1gH · 2025-11-01

**Soundness:** 3
**Presentation:** 2
**Contribution:** 2
**Rating:** 4
**Confidence:** 4

**Summary:**

This paper introduces a dynamic hair reconstruction method from multi-view videos. The hair geometry is represented by explicit 3D strands, and its appearance is modeled by a set of strand-aligned 3D Gaussians. To accommodate dynamic settings, the authors propose to use K-Plane representation to factorize the U x V x T x D feature volume into three planes across the spatial and temporal dimensions, which are later optimized in a coarse-to-fine manner. Several optimization acceleration techniques are introduced as well, resulting in speed up in the training. Experiments show better performance compared to baseline methods (mainly for dynamic hair reconstruction).

**Strengths:**

Hair reconstruction is hard, especially in the dynamic settings. This paper shows decent reconstruction quality on the shown examples, where strand-level details are sharper compared to other baselines in the supplemental video.

**Weaknesses:**

I feel this paper is written in a hurry so there are some details remaining unclear and unconvincing to me in the manuscript, as well as some easy-to-fix typos (e.g., paper title). I will list my points below.

[K-Plane]
- This paper uses K-Plane to represent spatio-temporal hair features, which is proven to be better than simply stacking all uv textures along the time dimension (shown in ablation study). One thing that is not verified is the neural network used to process it. The authors argue that MLP architectures work better than UNet in lines 232 - 235, yet there are no experiments to prove this point. Also, it’s unclear to me how the MLP is used to model K-Planes. Is it like a neural implicit representation that MLP approximates a 2D function (e.g., f(u, v))?

[Technical Details]
- In line 242 - 244, it seems the appearance output includes an additional confidence value, for which I don’t know the meaning and where it’s gonna be used. Is it just a different name for opacity (as I don’t see opacity output in appearance)?

[Results]
- The currently provided results are quite limited, only on a few identities, and the supplemental video even only includes one example. I feel it's insufficient to evaluate the performance of this work, especially for a dynamic reconstruction work but only one example is provided in the video. I would expect to see the (dynamic) reconstruction results on more diverse hairstyles, such as those more challenging curly hairstyles.
- For strand-based hair reconstruction, I think geometry is more important than appearance as high-quality hair geometry has a wider application in down-stream tasks. Regarding the current manuscript, first it’s unclear to me which metrics are geometry metrics and which are appearance metrics. To me, all metrics (PSNR, SSIM, LPIPS) measure the final rendering quality and don’t focus on the geometry quality only. Please correct me if I’m wrong. Second, I would expect more strand visualization as the one shown in the teaser, as it would provide clearer inspection on the geometric details.

**Questions:**

I list most of my questions in the Weaknesses section. Regarding them, I will put my current score as borderline reject, and I’m looking forward to the rebuttal phase to see if these concerns can be addressed.

---

### Note · Authors · 2025-11-13

I have read and agree with the venue's withdrawal policy on behalf of myself and my co-authors.